# Antioxidant, Neuroprotective, and Antinociceptive Effects of Peruvian Black Maca (*Lepidium meyenii* Walp.)

**DOI:** 10.3390/antiox14101214

**Published:** 2025-10-08

**Authors:** Iván M. Quispe-Díaz, Roberto O. Ybañez-Julca, Daniel Asunción-Alvarez, Cinthya Enriquez-Lara, José L. Polo-Bardales, Rafael Jara-Aguilar, Edmundo A. Venegas-Casanova, Ricardo D. D. G. de Albuquerque, Noé Costilla-Sánchez, Edison Vásquez-Corales, Pedro Buc Calderon, Julio Benites

**Affiliations:** 1Grupo de Investigación en Estudios de Compuestos Naturales y Sintéticos con Actividad a Nivel Sistema Nervioso Central y Musculo Liso, Laboratorio de Farmacología, Facultad de Farmacia y Bioquímica, Universidad Nacional de Trujillo, Trujillo 13011, Peru; iquispe@unitru.edu.pe (I.M.Q.-D.); hasuncion@unitru.edu.pe (D.A.-A.); 2Programa de Doctorado en Química Medicinal, Facultad de Ciencias de la Salud, Universidad Arturo Prat, Casilla 121, Iquique 1110939, Chile; cenriquez@estudiantesunap.cl (C.E.-L.); pedro.buccalderon@uclouvain.be (P.B.C.); 3Facultad de Farmacia y Bioquímica, Universidad Nacional de Trujillo, Trujillo 13011, Peru; jpolo@unitru.edu.pe (J.L.P.-B.); djara@unitru.edu.pe (R.J.-A.); evenegas@unitru.edu.pe (E.A.V.-C.); 4Laboratório de Tecnologia em Produtos Naturais, Universidade Federal Fluminense, Niterói 24020-140, RJ, Brazil; richardcabofrio@gmail.com; 5Laboratorio de Métodos Instrumentales, Facultad de Ingeniería Química, Universidad Nacional de Trujillo, Trujillo 13011, Peru; ncostilla@unitru.edu.pe; 6Escuela de Farmacia y Bioquímica, Universidad Católica Los Ángeles de Chimbote, Chimbote 02801, Peru; evasquezc@uladech.edu.pe; 7Research Group in Metabolism and Nutrition, Louvain Drug Research Institute, Université Catholique de Louvain, 73 Avenue E. Mounier, GTOX 7309, 1200 Brussels, Belgium; 8Laboratorio de Química Medicinal, Química y Farmacia, Facultad de Ciencias de la Salud, Universidad Arturo Prat, Casilla 121, Iquique 1110939, Chile

**Keywords:** black maca, *Lepidium meyenii* Walp., antioxidant activity, neuroprotection, ovariectomized rats, antinociceptive effects

## Abstract

*Lepidium meyenii* Walp. (black maca, BM) is a traditional Andean crop increasingly studied for its bioactive potential. This work characterized the phytochemical profile and evaluated the antioxidant, antinociceptive, and neuroprotective properties of a lyophilized aqueous extract of BM hypocotyls. UHPLC-ESI-QTOF-MS/MS identified twelve major compounds, including macamides, imidazole alkaloids, sterols, and fatty acid amides. BM showed a moderate total phenolic content but strong electron transfer-based antioxidant activity in CUPRAC and FRAP assays, together with moderate radical scavenging capacity in ABTS and DPPH systems. In ovariectomized rats, BM significantly reduced brain malondialdehyde levels, mitigated oxidative stress, and improved spatial learning during acquisition in the Morris water maze, confirming its neuroprotective effect. Antinociceptive assays (hot plate, cold plate, and tail immersion) further revealed a rapid but transient increase in nociceptive thresholds. This study provides experimental evidence supporting the analgesic effect of black maca. Molecular docking highlighted lepidiline B and campesterol as key metabolites with strong interactions with redox enzymes, the μ-opioid receptor, and the FAAH enzyme, supporting their role in the observed bioactivities. ADMET predictions indicated favorable oral bioavailability, CNS penetration, systemic clearance, and acceptable safety profiles. These results substantiate the role of black maca as a neuroprotective nutraceutical and highlight its promise as a novel source of rapidly acting natural analgesic compounds.

## 1. Introduction

Over the past two decades, there has been a significant increase in global interest and demand for *Lepidium meyenii* Walp, commonly referred to as “maca.” Consequently, maca has become one of Peru’s leading products, marketed in multiple presentations including powders, capsules, pills, flour, liquor, and extracts and commercialized in retail establishments such as health food stores and smoothie shops [1].

Maca roots are native to the Peruvian Andes region and belong to the Brassicaceae family. They thrive in high-altitude regions characterized by harsh environmental conditions such as rocky soils, intense solar radiation, and strong winds [2,3,4,5,6]. The availability of maca hypocotyls in various colors highlights their potential therapeutic versatility, which has been associated with diverse pharmacological effects [7]. These include the regulation of sexual dysfunction [8], neuroprotective effects [9,10], memory enhancement, and antidepressant properties [8,11]. Moreover, maca root supplements are rich in antioxidants, and possess anticancer, anti-inflammatory, and photoprotective properties [12,13,14,15]. Nutritionally, dried maca hypocotyls contain high levels of carbohydrates, proteins, lipids, essential amino acids, and free fatty acids. In addition, several secondary metabolites have been identified, including macamides, macaridine, alkaloids, and glucosinolates [16].

Within Peru, the Department of Junín (Carhuamayo) has recognized 13 distinct maca varieties ranging in color from white to black. Consumption of these varieties has been linked to enhanced energy levels, improved attentiveness, and hormonal balance. Among them, black maca is the rarest, representing approximately 15% of the annual harvest, yet it is also considered the most effective variety for men, particularly with respect to muscle gain, endurance, cognitive performance, and libido [7].

Importantly, maca has been consistently associated with biological activities relevant to the central nervous system. It has been shown to enhance memory [11], exert antioxidant effects [17], improve cognitive function [18], inhibit acetylcholinesterase activity, and maintain unaltered monoamine oxidase levels in male mice. In this context, ovariectomy is widely used as a female animal model of cognitive decline, since it induces memory impairment associated with increased oxidative stress, cholinergic dysfunction, and alterations in monoamine oxidase activity [19].

Based on this background, the present study aimed to investigate new pharmacological properties of this maca species and to confirm some of them previously reported. To this end, the chemical composition of the lyophilized aqueous extract of black maca hypocotyls was investigated, its total phenolic content was quantified, and its in vitro antioxidant activities was assessed using cupric-reducing antioxidant power (CUPRAC), ferric-reducing antioxidant power (FRAP), 2,2′-azino-bis(3-ethylbenzothiazoline-6-sulfonic acid) (ABTS^•+^), and 2,2-diphenyl-1-picrylhydrazyl (DPPH) assays. Furthermore, the evaluation of the extract’s analgesic effects was performed in rat models, while its impact on spatial memory was conducted in ovariectomized rats. These findings were complemented with in silico molecular docking analyses targeting proteins involved in redox regulation, and endocannabinoid metabolism involved in pain modulation and neuroprotection. Finally, pharmacokinetic and toxicity-related properties were predicted using the pkCSM online tool https://biosig.lab.uq.edu.au/pkcsm/ (accessed on 28 September 2025).

## 2. Materials and Methods

### 2.1. Chemicals, Drugs, and Solvents

All the solvents and reagents were purchased from different companies, such as Aldrich (St. Louis, MO, USA), Merck (Peruana S. A, Ate, Lima, Peru), and Eli Lilly (Indianapolis, IN, USA), and were used as supplied.

### 2.2. Plant Material

Hypocotyls of *Lepidium meyenii* Walp. (black maca) were collected in September 2024 in the Carhuamayo Province of Junin District, at 4000 m above sea level in Valle Mantaro, Peru. The specimen was taxonomically identified at the “*Herbarium Truxillense* (HUT) de la Facultad de Ciencias Biológicas de la Universidad Nacional de Trujillo,” where a voucher sample was deposited for reference.

### 2.3. Preparation of Lyophilized Aqueous Extract of Black Maca (BM)

The aqueous extract of *Lepidium meyenii* Walp. (black maca) hypocotyls was prepared following a standardized procedure. Briefly, 500 g of dried hypocotyls was weighed and added to 1.5 L of preheated distilled water at 50 °C. The mixture was refluxed for 2 h, and the extraction was repeated twice with another aqueous solution under the same conditions. The combined extracts were filtered to remove insoluble residues and evaporated under reduced pressure. The concentrated extract was frozen at −80 °C (Arctiko freezer, Nashville, TN, USA) and lyophilized using a freeze-dryer (Labconco, Kansas City, MO, USA), yielding 72.5 g of dried extract (14.5% *w*/*w*). Lyophilized samples were stored at +4 °C until further use.

### 2.4. Chemical Identification by HPLC-ESI-QTOFMS/ MS

Lyophilized BM was dissolved in methanol (1 mg/mL) and subjected to chromatographic separation using a UPLC Rapid Resolution System (Waters/Micromass, Milford, MA, USA) equipped with a binary pump, a degasser, and an automatic injector. The separation was performed on a ZORBAX Eclipse Plus C-18 column (2.1 × 150 mm, 5 μm) (Agilent Technologies Inc., Santa Clara, CA, USA) with a flow rate of 0.4 mL/min and a 2 μL injection volume. The mobile phase consisted of water with 0.1% acetic acid (A) and acetonitrile (B) using the following gradient: 0.0–4.7 min, 3–30% B; 4.7–7.8 min, 30–50% B; 7.8–11 min, 50–90% B; 11–12.5 min, 90% B; and 12.5–14 min, 90–93% B. The column effluent was split with a T-valve, and 20 μL/min was introduced into a micrO-TOF-Q™ orthogonal Q-TOF mass spectrometer (micrOTOF-QTM, Bruker Daltonics, Madison, WI, USA) with an electrospray ionization (ESI) source. Analyses were carried out in positive ion mode (*m*/*z* 100–1000) under the following conditions: capillary voltage: 4500 V, end-plate offset: −500 V, charging voltage: 2000 V, drying gas temperature: 200 °C, drying gas flow: 10.0 mL/min, nebulizer gas pressure: 4 bar, collision energy: 50 eV, and nitrogen as collision gas. Data acquisition and processing were performed with Bruker Compass Data Analysis 4.2 software (Bruker Daltonics, Madison, WI, USA).

### 2.5. Total Phenolic Content (TPC)

The TPC of BM was determined by the Folin–Ciocalteu method adapted from Singleton and Rossi [20]. Briefly, 25 μL of diluted extract was mixed with 125 μL of the Folin–Ciocalteu reagent and incubated for 20 min. Then, 100 μL of sodium carbonate (7%) was added, and the solution was incubated at 45 °C for 10 min. The absorbance of the resulting blue solution was measured at 760 nm. Results were expressed as mg gallic acid equivalents (GAE)/mL of BM.

### 2.6. Antioxidant Capacity Assays

#### 2.6.1. Cupric Reducing Antioxidant Capacity (CUPRAC) and Ferric-Reducing Antioxidant Power (FRAP) Assays

CUPRAC and FRAP assays were performed following established methods with slight modifications [21]. For CUPRAC, ammonium acetate buffer (pH 7.0), CuCl_2_, and neocuproine were mixed in a 1:1:1 ratio. Then, 10 μL of standard/sample and 265 μL of H_2_O were added, followed by 30 min incubation in the dark. Absorbance was measured at 450 nm. For FRAP, the working solution consisted of acetate buffer (300 mM, pH 3.6), TPTZ (2,4,6-tripyridyl-S-triazine) solution (10 mM in 40 mM HCl), and FeCl_3_·6H_2_O (20 mM), prewarmed at 37 °C. A Trolox^®^ (Sigma-Aldrich-Fluka, St. Louis, MO, USA) calibration curve (0.05–0.5 mM) was prepared. BM extract (8 μL) was incubated with 200 μL FRAP solution for 30 min in the dark, and absorbance was measured at 593 nm. The results were expressed as mg of Trolox^®^ equivalents (TE)/mL of BM. A standard curve was prepared using the standard antioxidant Trolox^®^.

#### 2.6.2. ABTS^•+^ Free-Radical Scavenging Assay

ABTS^•+^ scavenging activity was assessed according to Re et al. [22]. Trolox^®^ (1 mg/mL in ethanol) was serially diluted (50–800 μM). Then, 10 μL of each dilution was mixed with 300 μL of the ABTS^•+^ solution, and absorbance was measured at 750 nm using a Fisherbrand AccuSkan GO UV/Vis Microplate Spectrophotometer (Hampton, VA, USA). The BM samples were analyzed under the same conditions. The % inhibition was plotted against concentration, and IC_50_ values were calculated.

#### 2.6.3. DPPH Free-Radical Scavenging Assay

DPPH radical scavenging activity was determined following a modified protocol reported in the literature [23]. Trolox^®^ (1 mg/mL in ethanol) was serially diluted (0.1–1 mM). Then, 20 μL of each dilution was combined with 300 μL of DPPH solution and incubated for 30 min at room temperature. Absorbance was measured at 517 nm using a Fisherbrand accuSkan GO UV/Vis Microplate Spectrophotometer (Hampton, VA, USA). For BM samples, 20 μL of extract replaced Trolox^®^. IC_50_ values (concentration required to scavenge 50% of radicals) were calculated. All assays were performed in triplicate.

### 2.7. Animals and Experimental Groups

All experiments were performed in accordance with the American Veterinary Medical Association (AVMA) guidelines and approved by the Ethics Committee of the Faculty of Pharmacy and Biochemistry, Universidad Nacional de Trujillo (COD.N_: P 012-19/CEIFYB).

For analgesic activity, adult male *Rattus norvegicus* (Holtzman strain) (10–12 weeks, 200–250 g) were used. For the ovariectomy (OVX)-induced estrogen depletion model, as well as for memory and lipid peroxidation studies, adult female rats of the same strain and age range were included. Animals were housed individually under controlled temperature (22–25 °C), 12 h light/dark cycle, with *ad libitum* access to water and standard chow (Molinorte S.A.C., Trujillo, Peru). Rats were randomly assigned to five groups (*n* = 7 per group): (a) naïve non-ovariectomized (no OVX) controls; (b) ovariectomized (OVX) controls receiving 0.9% NaCl; (c) OVX + estradiol valerate (200 μg/kg); (d) OVX + BM (0.5 g/kg); and (e) OVX + BM (2.0 g/kg). Treatments were administered orally for two months.

### 2.8. Ovariectomy Procedure

Three-month-old female rats were anesthetized with ketamine (110 mg/kg, i.p.) [24]. A ventral incision was made, and in OVX groups, the ovaries, oviducts, and upper fallopian tubes were removed. Muscles and skin were sutured, and animals recovered under standard conditions. Behavioral experiments began three months post-surgery.

### 2.9. Lipid Peroxidation Assay

Lipid peroxidation in brain tissue was determined using the thiobarbituric acid reactive substances (TBARS) assay, which quantifies malondialdehyde (MDA) levels, following modified protocols [25,26]. Briefly, rat brains were homogenized in phosphate buffer (20 mM, pH 7.4, 140 mM KCl), centrifuged, incubated at 37 °C, precipitated with trichloroacetic acid, and reacted with thiobarbituric acid. The organic phase was extracted with butanol/pyridine (15:1), centrifuged, and absorbance measured at 532 nm. Results were expressed as μmol MDA/g tissue.

### 2.10. Morris Water Maze (MWM) Experiment

Spatial memory was assessed using the MWM test [27]. Rats were trained for four days (four trials/day, 120 s max) to locate a hidden platform submerged 1 cm below the water. Escape latency was recorded. On day 5, a probe trial (platform removed) was conducted, measuring time spent in the target quadrant and platform crossings.

### 2.11. Hot/Cold Plate Test

Thermal sensitivity was measured using a hot/cold plate analgesiometer (Ugo Basile^®^, Gemonio, Italy), as previously described [28]. Male rats were habituated for 15 min at 25 °C. For the hot plate test, the surface was maintained at 55 ± 0.1 °C (cut-off time 10 s). For the cold plate test, the plate was set at 5 ± 0.1 °C (cut-off 60 s). Latency to paw lifting or shaking was recorded. Response latency was measured before treatment and at 15, 30, 45, 60, 90, and 120 min post-dosing.

The treatments administered to animals in each experimental test group followed the study design outlined in Table 1.

### 2.12. Tail Immersion Test

Analgesic activity was also assessed by the tail immersion test [29]. A 1 L beaker containing 900 mL water with 0.44% NaCl was maintained at 55–56 °C, stirred continuously for uniform heating. Response latency was measured before treatment and at 60, 120, 180, 240 and 300 min post-dosing. The animals assigned to different experimental groups were treated in accordance with the study design described in Table 2.

### 2.13. Molecular Docking

Docking studies were performed with selected BM compounds on protein targets including NADPH oxidase [30], xanthine oxidase [31], superoxide dismutase [32], μ-opioid receptor (PDB: 4DKL) [33], and fatty acid amide hydrolase [34] (Figure 1). Molecular docking was carried out using AutoDock (v 4.2.1, Scripps Research Institute, San Diego, CA, USA), AutoDock Vina (v 1.0.2 Scripps Research Institute, San Diego, CA, USA) [35], and the AutoDockTools package v 1.5.7 [36], following protocols describe by Minchán-Herrera et al. [37]. Validation was performed by re-docking co-crystallized ligands into their corresponding binding sites, yielding RMSD values < 2.0 Å, thereby confirming the reliability of the docking protocol. All other parameters were set to the AutoDock Vina default. Docking simulations were repeated 20 times with the search exhaustiveness parameter set to 100. The best binding energy values (kcal·mol^−1^) were selected for evaluation. Three-dimensional docking results were visualized using the Discovery Studio 3.1 (Accelrys, San Diego, CA, USA) molecular graphics system.

### 2.14. ADMET Prediction

The pkCSM online tool (http://biosig.unimelb.edu.au/pkcsm/prediction, accessed on 26 March 2025) [38] was used to predict absorption, distribution, metabolism, excretion, and toxicity (ADMET) properties of the bioactive components of BM.

### 2.15. Statistical Analysis

Statistical analysis was conducted using the GraphPad Prism 8.0.2 software (San Diego, CA, USA). The data are presented as mean ± SEM. One-way or Two-way analysis of variance (ANOVA) with Tukey’s post hoc test was used for data analysis. Statistical significance was set at *p* ≤ 0.05.

## 3. Results

### 3.1. Chemical Composition of Lyophilized Hypocotyl Aqueous Extract of Black Maca

One of the objectives of this study was to characterize the phytochemical composition of the aqueous extract obtained from lyophilized hypocotyls of black maca (*Lepidium meyenii* Walp.). For this purpose, high-performance liquid chromatography coupled with electrospray ionization quadrupole time-of-flight tandem mass spectrometry (HPLC-ESI-QTOF-MS/MS) was employed. Representative spectra of the detected compounds are provided in the Appendix A.

In total, twelve major compounds were identified (Table 3), whose corresponding mass spectra are also shown in the Appendix A. These metabolites included sucrose, N-benzyloctanamide, N-(3-methoxybenzyl)-(9Z,12Z)-octadecadienamide, 1-benzyl-2-propyl-4,5-dimethylimidazilium, lepidiline A, lepidiline B, lepidiline D, campesterol, pinellic acid, 1,3-dibenzyl-2-pentyl-4,5-dimethylimidazilium, N-octadecanamide, and N-benzyl-16-hydroxy-9-oxo-13E,15E-octadecadienamide (Figure 2).

### 3.2. Total Phenolic Content and Antioxidant Capacity of BM

Table 4 summarizes the total phenolic content (TPC) and antioxidant activities of BM. The TPC was expressed as gallic acid equivalents (GAE) per mL of extract. The analysis revealed a moderate phenolic content of 10.62 mg GAE/mL in the BM. The antioxidant potential of BM was further assessed using four complementary assays that capture distinct mechanisms of action: (1) cupric ion-reducing antioxidant capacity (CUPRAC), (2) ferric-reducing antioxidant power (FRAP), (3) ABTS^•+^ radical scavenging, and (4) DPPH radical scavenging.

In the CUPRAC assay, which measures the reduction of the Cu(II)–neocuproine complex to the Cu(I)–neocuproine chromophore, BM exhibited strong reducing capacity with a TEAC value of 14.06 mg/mL. This activity was markedly higher than that of the reference antioxidant quercetin (TEAC = 4.69 mg/mL).

Similarly, in the FRAP assay, which reflects the electron-donating ability of antioxidants through the reduction of the Fe^3+^–TPTZ complex to Fe^2+^–TPTZ, BM demonstrated substantial reducing power, with a TEAC value of 7.84 mg/mL, more than double that of quercetin (3.14 mg/mL).

The ABTS•^+^ assay, which evaluates the capacity of antioxidants to neutralize radical cations, showed that BM had an IC_50_ of 15.26 µg/mL. Although this value is higher than those of quercetin (IC_50_ = 0.09 µg/mL) and Trolox^®^ (IC_50_ = 0.20 µg/mL), it nonetheless indicates a relevant ability to scavenge hydrophilic radicals.

Finally, the DPPH assay, which measures hydrogen- or electron-donating capacity toward the DPPH radical, revealed that BM exhibited an IC_50_ of 36.82 µg/mL. This demonstrates considerable radical scavenging activity, supporting the overall antioxidant potential of the extract.

### 3.3. Spatial Memory and Brain Lipidic Peroxidation in Ovariectomized Rats

The effects of BM on spatial learning and memory were assessed using the Morris water maze. During the acquisition phase (days 1–4), escape latency to locate the hidden platform was measured (Figure 3A), while spatial memory retention was evaluated on day 5 through the probe test, recording the time spent in the target quadrant and the number of crossings over the former platform location.

On day 1, no significant differences were observed between the groups (ANOVA, *p* > 0.05), confirming comparable baseline performance (Figure 3B). By day 2 (Figure 3C), the OVX control group exhibited a marked increase in escape latency (248.6 ± 42.5 s) compared with the naïve group (54.0 ± 18.7 s; *p* < 0.05), confirming the learning impairment induced by ovariectomy. Notably, BM at 0.5 g/kg (85.0 ± 28.2 s; *p* < 0.05 vs. control) significantly reduced escape latency, whereas estradiol (99.8 ± 19.8 s; *p* > 0.05) and BM at 2.0 g/kg (127.5 ± 53.2 s; *p* > 0.05) did not. On day 3 (Figure 3D), both estradiol (74.5 ± 21.1 s; *p* < 0.05) and BM 0.5 g/kg (82.7 ± 15.8 s; *p* < 0.05) significantly improved escape latency relative to controls (244.0 ± 46.7 s), while the 2.0 g/kg dose remained ineffective (116.8 ± 52.9 s; *p* > 0.05). On day 4 (Figure 3E), BM at 0.5 g/kg (22.3 ± 5.3 s; *p* < 0.05) and estradiol (40.0 ± 10.4 s; *p* < 0.05) maintained significantly reduced latencies compared to controls (202.0 ± 47.4 s), suggesting enhanced learning and consolidation. The BM 2.0 g/kg group (101.2 ± 47.9 s) again did not reach statistical significance. In the probe test (day 5), only estradiol-treated rats displayed significant improvements, with greater time in the target quadrant (32.8 ± 0.5 s vs. control: 23.0 ± 0.4 s; *p* < 0.05; Figure 3F) and increased platform crossings (6.20 ± 0.37 vs. control: 3.43 ± 0.78; *p* < 0.05; Figure 3G). Neither BM dose significantly improved probe test performance.

Given that lipid peroxidation is a free radical–driven process in which reactive oxygen species (ROS) attack polyunsaturated fatty acids in cellular membranes, we measured malondialdehyde (MDA) levels as a reliable biomarker of oxidative damage in neural tissue. In addition to behavioral outcomes, OVX animals exhibited a marked increase in brain MDA concentrations (3.44 ± 0.1 µmol/g tissue) compared with naïve rats (0.98 ± 0.08 µmol/g tissue; *p* < 0.001) (Figure 4). Estradiol treatment significantly reduced MDA concentrations (1.12 ± 0.07 µmol/g tissue; *p* < 0.001 vs. control), restoring values to those of naïve animals. Similarly, BM at both 0.5 g/kg (0.79 ± 0.08 µmol/g tissue) and 2.0 g/kg (1.06 ± 0.05 µmol/g tissue) significantly decreased MDA levels compared with OVX controls (*p* < 0.001). No significant differences were observed between the BM 2.0 g/kg and estradiol groups (*p* > 0.05), suggesting a comparable antioxidant effect. Although the 2.0 g/kg dose showed slightly higher MDA values than 0.5 g/kg, the difference was not statistically significant, indicating a plateau in the antioxidant efficacy of BM at higher concentrations.

### 3.4. Antinociceptive Assays

The antinociceptive effects of BM were first evaluated using the hot plate test following oral administration at doses of 0.25, 0.5, and 1.0 mg/kg. As shown in Figure 5, all doses of BM increased the reaction latency compared with the vehicle control group (0.9% NaCl), with effects evident as early as 15 min and peaking at 45 min. At 15 min, BM 1.0 mg/kg significantly prolonged reaction time by 67.2 ± 13.4% (*p* < 0.05 vs. control). The maximal effect occurred at 45 min, where BM 1.0 mg/kg produced the strongest response (108.2 ± 24.9%; *p* < 0.01), followed by BM 0.5 mg/kg (107.4 ± 17.3%; *p* < 0.05). At the same time point, tramadol (10 mg/kg) also increased response latency (63.6 ± 13.5%; *p* < 0.05). Between 15 and 45 min, all BM-treated groups (0.25–1.0 mg/kg) showed greater response latencies than tramadol, although the differences did not reach statistical significance. At 60 min, BM 0.5 mg/kg (71.8 ± 14.7%; *p* < 0.05) and BM 1.0 mg/kg (67.8 ± 8.2%; *p* < 0.01) maintained significant activity, while tramadol produced the highest latency (117.6 ± 11.7%; *p* < 0.001), not significantly different from BM. Interestingly, BM displayed an early onset of action (15–45 min) with transient efficacy, whereas tramadol produced a delayed but sustained response, reaching its maximum effect at 120 min (247.6 ± 45.4%; *p* < 0.05). After 90 min, BM-treated animals no longer exhibited significant increases in latency (*p* > 0.05).

The cold plate test further confirmed the antinociceptive potential of BM (Figure 6). All doses significantly increased reaction latency compared with controls, with effects observed at 15 min after administration. At this time point, BM 0.25 mg/kg increased latency by 48.3 ± 2.5% (*p* < 0.01 vs. control), while tramadol elicited a comparable effect (56.7 ± 9.6%; *p* < 0.01). At 30 min, tramadol produced the greatest response (88.5 ± 9.6%; *p* < 0.001), followed by BM 1.0 mg/kg (80.2 ± 9.5%; *p* < 0.001), BM 0.5 mg/kg (79.2 ± 8.4%; *p* < 0.001), and BM 0.25 mg/kg (60.5 ± 8.3%; *p* < 0.001). No statistical differences were found between BM-treated groups and tramadol at this time point (*p* > 0.05). The maximum response was recorded at 60 min in the tramadol group (168.3 ± 13.6%; *p* < 0.001), followed by BM 0.25 mg/kg (131.0 ± 18.1%; *p* < 0.001) and BM 0.5 mg/kg (51.0 ± 10.1%; *p* < 0.001). Interestingly, BM 1.0 mg/kg failed to produce a significant effect at 60 min. By 90 min, efficacy declined in all groups, although tramadol and BM (0.25 and 0.5 mg/kg) still showed higher latencies than controls (*p* < 0.05). At 120 min, only tramadol maintained significant activity (36.3 9.0%; *p* < 0.05).

The tail immersion test provided additional evidence of BM’s antinociceptive activity (Figure 7). Oral administration of BM at 0.25, 0.5, and 1.0 mg/kg significantly increased tail-flick latency compared with vehicle-treated rats, with effects evident at 60 min post-treatment. The maximal response was observed at this time with BM 1.0 mg/kg (174.8 ± 23.2%; *p* < 0.01 vs. control), followed by BM 0.5 mg/kg (81.2 ± 12.3%; *p* < 0.01). Notably, BM 1.0 mg/kg was significantly more effective than tramadol (29.5 ± 12.7%; *p* < 0.01). At 120 min, all BM doses remained significantly active (BM 1.0 mg/kg: 82.5 ± 16.7%; BM 0.5 mg/kg: 132.7 ± 18.1%; BM 0.25 mg/kg: 86.5 ± 21.6%; *p* < 0.05–0.01), with no differences compared to tramadol (97.2 ± 21.9%). By 180 min, only tramadol-treated animals maintained significant activity (114.8 ± 20.5% vs. control: 20.0 ± 10.3%; *p* < 0.01). At 240 and 300 min, no significant antinociceptive effects were observed in any of the groups.

### 3.5. Molecular Docking

Molecular docking analysis was performed to investigate the potential interactions of black maca constituents with key protein targets relevant to redox regulation, analgesia, and neuroprotection. The selected targets included NADPH oxidase (PDB: 2CDU), xanthine oxidase (PDB: 3NRZ), and superoxide dismutase (PDB: 4MCM) as representative redox enzymes; the μ-opioid receptor (PDB: 4DKL) as an analgesic target; and fatty acid amide hydrolase (FAAH, PDB: 2VYA), a central enzyme in endocannabinoid metabolism involved in pain modulation and neuroprotection.

As summarized in Table 5, several compounds demonstrated strong binding affinity across multiple targets. Among the redox enzymes, campesterol (compound **8**) exhibited the most robust docking scores, particularly with ROS-generating enzymes such as NADPH oxidase (−9.9 kcal/mol) and xanthine oxidase (−9.5 kcal/mol), suggesting strong inhibitory potential. Consequently, a low binding activity was observed with the antioxidant enzyme SOD (−6.6 kcal/mol). Similarly, lepidiline B (compound **6**), lepidiline D (compound **7**), and 1,3-dibenzyl-2-pentyl-4,5-dimethylimidazilium (compound **10**) consistently displayed high binding affinities (−8.7 to −9.6 kcal/mol), reinforcing their role as redox candidates.

Docking results for the μ-opioid receptor further highlighted campesterol (−8.6 kcal/mol) as one of the top ligands, surpassing tramadol (−6.4 kcal/mol), which was used as a reference compound. Lepidilines (compounds **5**–**7**) also showed strong interactions (−8.2 to −8.4 kcal/mol), supporting their potential contribution to the analgesic effects observed in vivo.

For FAAH, compound **6** displayed the strongest binding affinity (−10.1 kcal/mol), followed by compound **8** (−9.7 kcal/mol) and compound **7** (−9.3 kcal/mol). These values were comparable to estradiol (−9.4 kcal/mol), used as a reference ligand.

The docking interactions (Figure 8) illustrate the binding modes of lepidiline B (compound **6**) and campesterol (compound **8**) with key protein targets related to redox homeostasis, nociception, and neuroprotection.

The stability of these complexes was mainly mediated by hydrogen bonds, π–sulfur interactions, amide–π stacked, and π–π or π–σ stacking interactions, which collectively compensated for less favorable donor–donor contacts. In NADPH oxidase (Figure 8A), compound **6** formed stable interactions with residues such as Leu40, Pro432, Arg431, and Tyr62. In contrast compound **8** (Figure 8A1) engaged a broader set, including His10, Phe14, Gly59, Tyr62, Val304, Phe367, Arg431, Pro432, and Phe433, with an additional hydrogen bond to Gly59 that further stabilized the complex. In xanthine oxidase, compound **6** (Figure 8B) primarily interacted with Ala301, Ala302, Ile353, Leu287, Val259, Pro281, Ile403, and Leu257 whereas compound **8** (Figure 8B1) established network of contacts, including Ile353, Leu257 and Ala301, consistent with stronger binding stability. At the μ-opioid receptor, compound **6** (Figure 8C) displayed π–σ stacking interactions with Val236, Ile296 and Ile322, together with π–σ interactions involving Val300. Additional π–alkyl interaction with Tyr326 and Trp293, along with π–sulfur interactions with Met151, further stabilized the complex. Compound **8** (Figure 8C1) engaged a broader set of residues, including Val236, Met151, Ile296, Ile322, Tyr128, Trp133 and Val300, reinforcing its high docking score. For FAAH, both compounds (Figure 8D,D1) formed a complex network of hydrogen bonding, and π–π stacking and π–δ interactions, consistent with the strong docking energies reported in Table 6. Detailed residue-level interaction profiles (Appendix A) and the superimposition of experimental and re-docked ligand poses (Appendix A) are provided in the Appendix A.

### 3.6. ADMET Profiles of BM Metabolites

Pharmacokinetic and toxicity parameters were predicted using the pkCSM online tool, and the results are summarized in Table 6.

Overall, all compounds complied with Lipinski’s rule of five, as their molecular weights were below 500 g/mol, supporting their predicted oral bioavailability. Caco-2 permeability values ranged from −0.288 to 1.630 log Papp (10^−6^ cm/s). Compounds **1** and **9** exhibited the lowest permeability (−0.288 and 0.861, respectively), whereas compound **2** displayed the highest value (1.630). Consistently, intestinal absorption (IA) exceeded 87% for most compounds, except for compounds **1** (2.57%) and **9** (38.81%), which showed markedly reduced absorption and therefore compromised oral bioavailability. In contrast, skin permeability was uniformly low (log Kp between −2.206 and −2.836 cm/h). Only compound **2** (−2.206) approached the threshold for potential dermal absorption, while the remaining compounds were unlikely to achieve meaningful transdermal penetration.

Regarding distribution, the predicted steady-state volumes of distribution (VDss) ranged from −0.882 to 1.637 log L/kg. Compounds **6** and **10** exhibited the highest values, suggesting broad systemic distribution. Blood-brain barrier (BBB) permeability values (log BB) ranged from −1.697 to 1.108. Notably, several compounds exceeded the threshold of 0.3, indicating the potential for central nervous system penetration, which may contribute to their neuroprotective activity.

With respect to metabolism, most compounds were not predicted to inhibit cytochrome P450 isoforms, suggesting a low likelihood of drug-drug interactions. Nevertheless, compounds **5**, **6**, **7**, and **10** were predicted to inhibit CYP2D6, while compounds **7** and **12** inhibited CYP3A4, highlighting a possible risk of metabolism-based interactions that warrants further investigation.

Excretion analysis indicated total clearance values ranging from 0.572 (compound 8) to 1.990 (compound **12**), consistent with efficient systemic elimination. Finally, acute oral toxicity (rat LD_50_) values ranged from 0.565 to 3.265 mol/kg, with compound **6** predicted as the most toxic and compound **12** as the least toxic.

## 4. Discussion

The chemical composition analysis of the lyophilized aqueous extract of black maca revealed a profile consistent with previous reports, confirming the presence of macamides, imidazole alkaloids, phytosterols, and fatty acid amides with potential biological relevance [46,47]. These constituents encompass a wide range of bioactive properties. For instance, macamides such as N-benzyloctanamide and its derivatives have been linked to neuroprotective, anti-inflammatory, and energizing effects, likely mediated through endocannabinoid modulation [48,49]. Their detection provides a plausible mechanistic basis for some of the cognitive and behavioral benefits traditionally attributed to this plant. Likewise, campesterol, a representative phytosterol, has been associated with cholesterol-lowering and anti-inflammatory activities [50], reinforcing the potential role of black maca in metabolic regulation and cardiovascular health. The identification of imidazole alkaloids, including lepidilines A, B, and D, further supports the hypothesis that black maca exerts adaptogenic and neuroactive effects, consistent with its ethnopharmacological use in Andean populations [51,52]. In addition, fatty acid amides such as N-octadecanamide and hydroxylated derivatives may contribute to neural signaling and energy metabolism [53].

BM exhibited a moderate total phenolic content (TPC), consistent with the variable accumulation of polyphenolic compounds in maca hypocotyls, which is known to be influenced by biosynthetic processes stimulated by environmental factors such as light exposure [54]. In antioxidant assays, BM showed strong activity in electron-transfer–based methods (CUPRAC and FRAP), indicating the presence of metabolites capable of stabilizing oxidants through single-electron donation [55]. In contrast, relatively high IC_50_ values in ABTS and DPPH assays indicated lower efficiency in direct radical scavenging. This profile may reflect both the chemical nature of the extract—imidazole alkaloids and macamides likely account for its reducing power, whereas the limited abundance of phenolic hydroxyl groups restricts hydrogen atom transfer [56,57]—and the aqueous matrix, which could limit the solubility and reactivity of lipophilic compounds effective in DPPH-type systems.

With respect to neurocognitive outcomes, the Morris water maze confirmed that OVX rats developed deficits in spatial learning and memory, consistent with hippocampal dysfunction and cholinergic decline in estrogen-deficient models [58,59,60]. As expected, estradiol treatment reversed these impairments. Interestingly, BM at 0.5 g/kg reproduced the cognitive benefits of estradiol during the acquisition phase, suggesting a role in learning acquisition and consolidation. However, BM did not improve performance in the probe test, indicating a limited impact on memory retrieval. Previous studies in mice have reported that black maca improves memory impairment in ovariectomized models, an effect associated with antioxidant activity and acetylcholinesterase inhibition [19]. In addition, another study demonstrated that black maca exerted the strongest benefits on latent learning when compared with red and yellow maca [11]. More recently, it has also been shown that maca enhances cognitive function in middle-aged mice, an effect linked to improved mitochondrial activity and the upregulation of autophagy-related proteins in the cortex [18]. In the present study, the absence of efficacy at 2.0 g/kg suggests paradoxical dose-dependent responses, possibly related to receptor desensitization or metabolic saturation. Because lipid peroxidation is a key mechanism of oxidative damage in the brain, we measured malondialdehyde (MDA) levels—a widely used biomarker of lipid peroxidation in ovariectomized animal models. Previous studies in OVX models have shown that elevated brain MDA concentrations are associated with cognitive impairments, and that both parameters can be attenuated by antioxidant interventions [61,62]. Biochemically, BM significantly reduced brain malondialdehyde (MDA) levels in OVX rats, reaching values comparable to estradiol. Notably, estradiol has been shown to act through the Nrf2 pathway in similar models [63]. Therefore, it is plausible that BM may also exert part of its antioxidant effects via Keap1–Nrf2 signaling, a hypothesis that warrants further investigation.

The antinociceptive assays provided complementary evidence of BM’s pharmacological activity. In the hot plate test, BM produced a dose-dependent analgesic effect with rapid onset and peak activity at 45 min. This earlier onset compared with tramadol, a μ-opioid receptor agonist, suggests the participation of non-opioid central pathways, potentially involving endocannabinoid, monoaminergic, or nitric oxide signaling [38,64]. The cold plate test corroborated the rapid onset but also revealed a transient efficacy, with a possible ceiling effect at higher doses. Similarly, the tail immersion test demonstrated robust activity at 1.0 g/kg, peaking at 1–2 h but declining after 180 min, indicating rapid metabolism or clearance of active constituents. Overall, BM exhibited a pharmacological profile characterized by fast-acting but short-lived analgesia, which could be advantageous in acute pain management. Previous studies have reported that maca alleviates articular and neuropathic pain by reducing mechanical hypersensitivity and postural imbalance in monoiodoacetate and sciatic nerve injury models [65]. In addition, macamides from maca have been identified as potent inhibitors of soluble epoxide hydrolase, and specific compounds such as N-benzyl-linoleamide have been shown to reduce inflammatory pain in vivo [66]. These findings are consistent with its phytochemical composition of BM, particularly macamides, lepidilines, and fatty acid amides, which are known to modulate endocannabinoid and monoaminergic pathways [67].

Molecular docking analysis provided further mechanistic insights. Compounds containing imidazilium or benzylamide scaffolds (e.g., lepidilines) displayed high binding affinities across multiple targets, while campesterol showed strong interactions consistent with its membrane-stabilizing and antioxidant properties. Notably, both lepidiline B and campesterol formed extensive stabilizing interactions with FAAH, suggesting a role in endocannabinoid modulation that could contribute to both analgesic and neuroprotective effects. This multitarget binding profile highlights their potential as lead scaffolds for the development of novel antioxidant and neuroactive agents. Molecular docking suggests that lepidiline B and campesterol may contribute to the observed effects, these compounds however were not directly tested. Future studies with isolated standards are needed to confirm their roles and possible synergistic interactions within the black maca extract.

The ADMET analysis revealed generally favorable pharmacokinetic and safety profiles. All compounds complied with Lipinski’s rule of five, with high intestinal absorption predicted for most molecules [68,69]. Skin permeability was limited, consistent with restricted transdermal diffusion [70], while several compounds displayed BBB permeability above the 0.3 threshold, suggesting potential CNS penetration [71]. Most compounds were not predicted to inhibit CYP450 isoforms; however some showed inhibition of CYP2D6 and CYP3A4. Such interactions may pose potential risks of altered drug metabolism, including reduced clearance, increased plasma concentrations, and enhanced toxicity. Predicted systemic clearance was efficient, and in silico acute oral toxicity values were within acceptable ranges according to OECD guidelines [72]. Among the tested metabolites, compounds **2**, **3**, **5**, and **12** emerged as the most promising candidates, combining efficient absorption, wide distribution, acceptable clearance, and relatively low toxicity.

Taken together, these findings indicate that the bioactivity of black maca is likely mediated through a polypharmacological mechanism involving antioxidant, neuroprotective, and analgesic pathways. Although molecular docking shows that lepidiline B and campesterol may contribute to the observed effects, these metabolites were not directly tested in the bioassays. This constitutes a limitation of the present study, and future research should address the isolation and experimental assessment of these compounds to clarify their individual contributions as well as their potential synergistic effects within the extract.

## 5. Conclusions

Phytochemical profiling of black maca revealed a characteristic composition rich in macamides, imidazole alkaloids, sterols, and fatty acid derivatives. The extract demonstrated strong antioxidant activity through electron transfer mechanisms and moderate radical scavenging capacity. In vivo, BM improved spatial learning and reduced lipid peroxidation in brain tissue of ovariectomized rats, consolidating its neuroprotective role. In parallel, antinociceptive assays provided experimental evidence supporting its fast-acting but short-lasting analgesic effects. Docking studies identified lepidiline B and campesterol as lead candidates with multitarget interactions across redox enzymes, the μ-opioid receptor, and the FAAH enzyme. ADMET predictions supported their drug-likeness with favorable bioavailability and safety. Collectively, these findings reinforce the scientific basis for the neuroprotective effects of black maca and highlight its potential as a natural analgesic candidate. Future investigations should assess isolated metabolites to validate their specific contributions and to elucidate potential synergistic mechanisms.

## Figures and Tables

**Figure 1 antioxidants-14-01214-f001:**
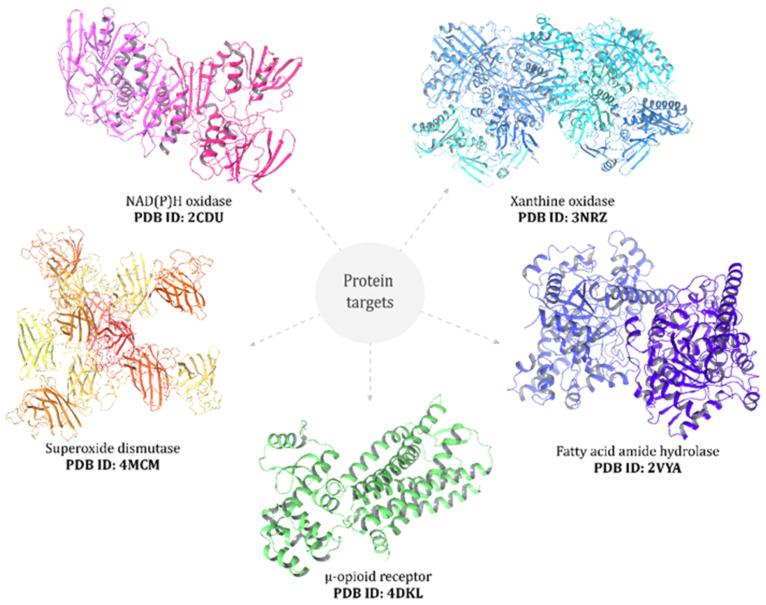
Black maca extract was docked into the binding sites of selected protein targets: NAD(P)H oxidase, xanthine oxidase, superoxide dismutase, μ-opioid receptor, and fatty acid amide hydrolase (PDB IDs indicated in the figure).

**Figure 2 antioxidants-14-01214-f002:**
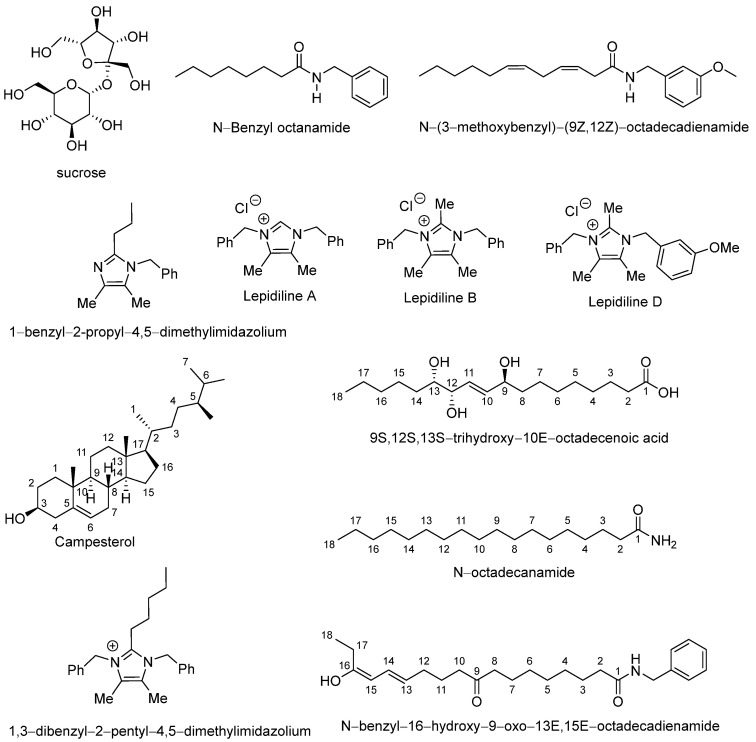
Chemical structures of abundant compounds identified in the lyophilized hypocotyl aqueous extract of black maca (*Lepidium meyenii* Walp.).

**Figure 3 antioxidants-14-01214-f003:**
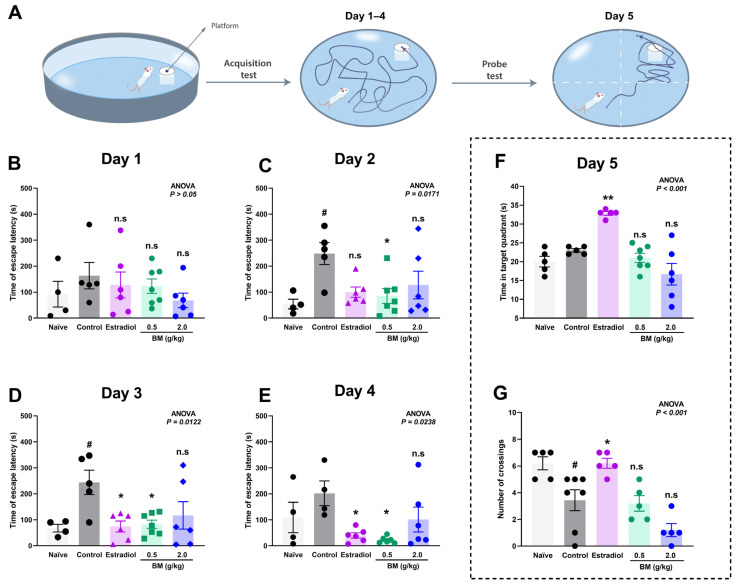
Effect of the lyophilized aqueous extract of black maca (*Lepidium meyenii* Walp., BM) on spatial learning and memory performance in ovariectomized rats assessed using the Morris water maze. (**A**) Experimental design. Escape latency time (s) to find the hidden platform was recorded during the acquisition phase: (**B**) day 1, (**C**) day 2, (**D**) day 3, and (**E**) day 4. On day 5, memory retention was evaluated by measuring (**F**) the time spent in the target quadrant and (**G**) the number of crossings over the former platform location. Data are expressed as mean ± SEM (*n* = 7 per group). Statistical significance was determined by one-way ANOVA followed by Tukey’s post hoc test. # *p* < 0.05 vs. naïve group; * *p* < 0.05, ** *p* < 0.01 vs. control group; n.s., not significant vs. control group.

**Figure 4 antioxidants-14-01214-f004:**
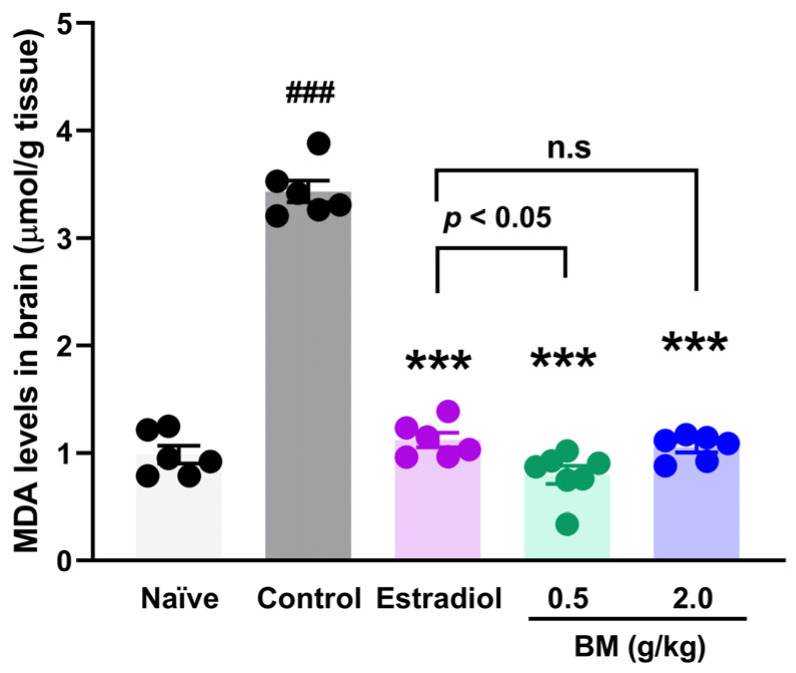
Effect of the lyophilized aqueous extract of black maca (*Lepidium meyenii* Walp., BM) on brain lipid peroxidation, assessed by malondialdehyde (MDA) levels, in ovariectomized rats. Data are expressed as mean ± SEM (*n* = 7 per group). Statistical significance was determined by one-way ANOVA followed by Tukey’s post hoc test. *** *p* < 0.001 vs. control group; ### *p* < 0.001 vs. naïve group; n.s., not significant vs. estradiol group.

**Figure 5 antioxidants-14-01214-f005:**
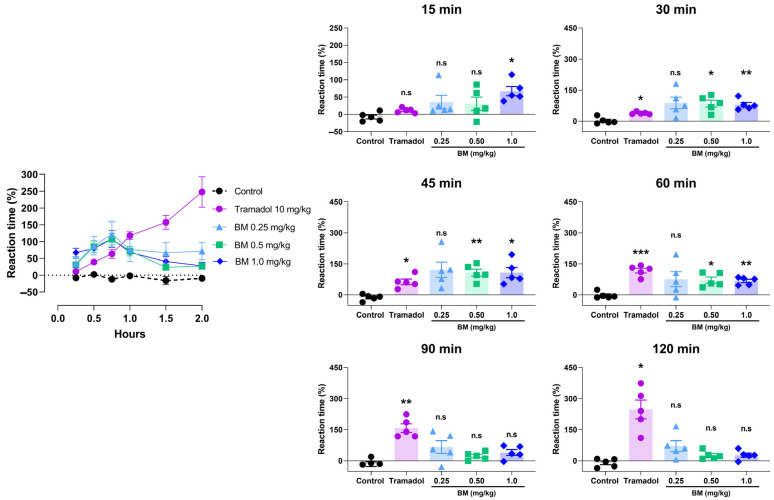
Antinociceptive effect of the lyophilized aqueous extract of black maca (*Lepidium meyenii* Walp., BM) hypocotyl in the hot-plate test in rats. Reaction latency (time to paw licking or jumping) was recorded at baseline (0 min) and at 15, 30, 45, 60, 90, and 120 min after oral administration of BM at doses of 0.25, 0.5, and 1.0 mg/kg. Tramadol (10 mg/kg, p.o.) was used as the reference drug, and 0.9% NaCl served as the vehicle control. Data are expressed as mean ± SEM (*n* = 6 per group). Statistical significance was determined by two-way ANOVA followed by Tukey’s post hoc test. * *p* < 0.05, ** *p* < 0.01, *** *p* < 0.001 vs. control group; n.s., not significant vs. control group.

**Figure 6 antioxidants-14-01214-f006:**
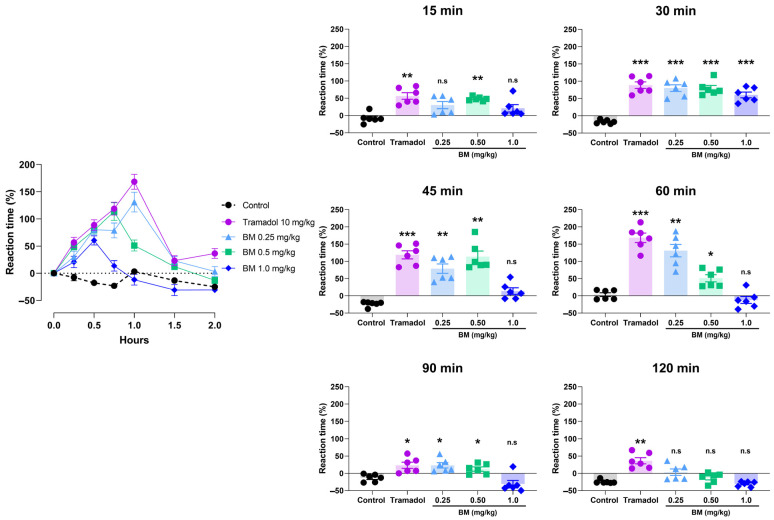
Antinociceptive effect of the lyophilized aqueous extract of black maca (*Lepidium meyenii* Walp., BM) hypocotyl in the cold-plate test in rats. Reaction latency (time to paw licking or jumping) was recorded at baseline (0 min) and at 15, 30, 45, 60, 90, and 120 min after oral administration of BM at doses of 0.25, 0.5, and 1.0 mg/kg. Tramadol (10 mg/kg, p.o.) was used as the reference drug, and 0.9% NaCl served as the vehicle control. Data are expressed as mean ± SEM (*n* = 6 per group). Statistical significance was determined by two-way ANOVA followed by Tukey’s post hoc test. * *p* < 0.05, ** *p* < 0.01, *** *p* < 0.001 vs. control group; n.s., not significant vs. control group.

**Figure 7 antioxidants-14-01214-f007:**
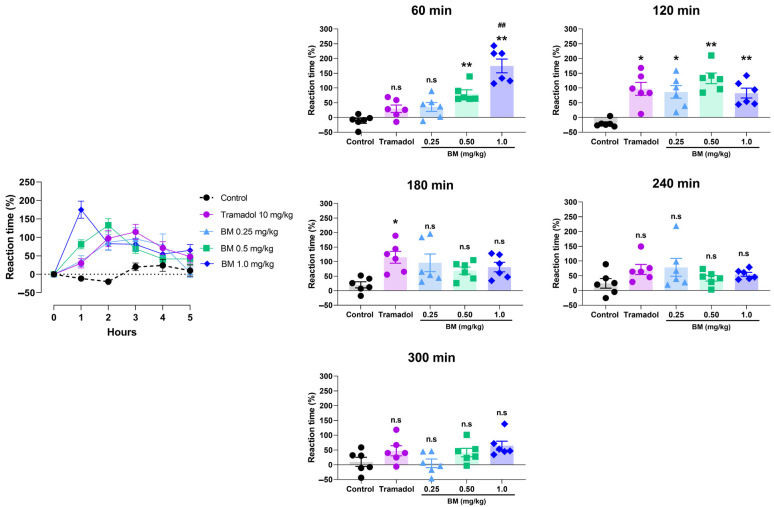
Antinociceptive effect of the lyophilized aqueous extract of black maca (*Lepidium meyenii* Walp., BM) hypocotyl in the tail immersion test in rats. Tail-flick latency was recorded at baseline (0 h) and at 60, 120, 180, 240, and 300 min. after oral administration of BM at doses of 0.25, 0.5, and 1.0 mg/kg. Tramadol (10 mg/kg, p.o.) was used as the reference drug, and 0.9% NaCl served as the vehicle control. Data are expressed as mean ± SEM (*n* = 6 per group). Statistical significance was determined by two-way ANOVA followed by Tukey’s post hoc test. * *p* < 0.05, ** *p* < 0.01 vs. vehicle; ## *p* < 0.01 vs. tramadol; n.s., not significant vs. control group.

**Figure 8 antioxidants-14-01214-f008:**
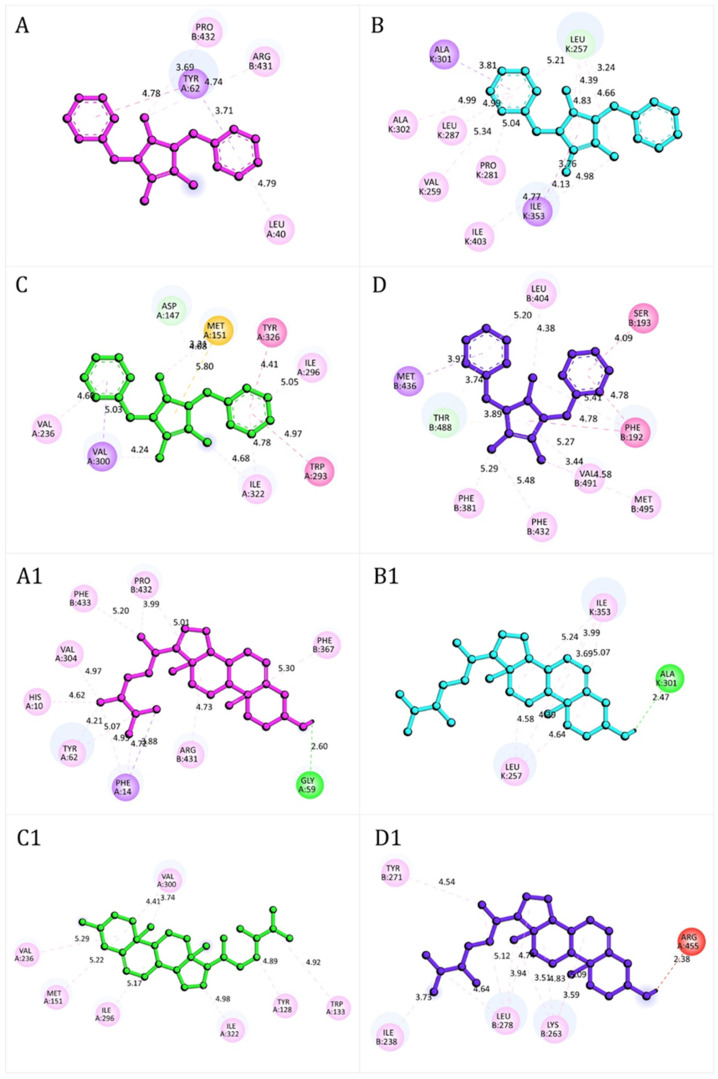
Representative docking interactions of compound **6** (**A**–**D**) and compound **8** (**A1**–**D1**) with NADPH oxidase (**A**,**A1**), xanthine oxidase (**B**,**B1**), μ-opioid receptor (**C**,**C1**), and fatty acid amide hydrolase receptor (**D**,**D1**). Hydrogen atoms are omitted in some cases for clarity. Interaction types: green, conventional hydrogen bonds; light green, carbon-hydrogen bonds or π-donor hydrogen bonds; red, unfavorable donor–donor contacts; purple, π (aromatic)–σ interactions; yellow, π–sulfur interactions; fuchsia, amide–π stacked or π–π T-shaped interactions; pink, alkyl or π–alkyl interactions.

**Table 1 antioxidants-14-01214-t001:** Experimental design of the hot/cold plate test in rats.

Groups	Treatment Allocation	Oral Route Dose
Control	NaCl 0.9%	0.1 mL/100 g
Reference standard	tramadol	10 mg/kg
Test group I	1–6	0.25 mg/kg
Test group II	7–12	0.50 mg/kg
Test group III	13–18	1.0 mg/kg

**Table 2 antioxidants-14-01214-t002:** Experimental design of the tail immersion test in rats.

Groups	Treatment Allocation	Oral Route Dose
Control	NaCl 0.9%	0.1 mL/100 g
Reference standard	tramadol	10 mg/kg
Test group I	1–6	0.25 mg/kg
Test group II	7–12	0.50 mg/kg
Test group III	13–18	1.0 mg/kg

**Table 3 antioxidants-14-01214-t003:** Chemical composition of lyophilized hypocotyl aqueous extract of black maca (*Lepidium meyenii* Walp.) determined by HPLC-ESI-QTOF-MS/MS.

N°	Proposed Compound	Molecular Formula	RT (min)	Mode of Ionization		Main Fragments (*m*/*z*)	References
				[M + H]	[M − H]		
1	Sucrose	C_12_H_22_O_11_	0.8	-	341	179	[39]
2	N-benzyloctanamide	C_15_H_23_NO	1.4	270	-	202	[40]
3	N-(3-methoxybenzyl)-(9Z,12Z)-octadecadienamide	C_26_H_41_NO_2_	4.8	399	-	292, 239, 150	[41]
4	1-benzyl-2-propyl-4,5-dimethylimidazilium	C_15_H_20_N_2_	10.7	227	-	186	[42]
5	1,3-dibenzyl-4,5-dimethylimidazilium chloride (Lepidiline A)	C_19_H_21_ClN_2_	17.3	277	-	239	[43]
6	1,3-dibenzyl-2,4,5-trimethylimidazilium chloride (Lepidiline B)	C_20_H_23_ClN_2_	17.9	291	-	239	[42]
7	3-benzyl-1-(3-methoxybenzyl)-2,4,5-trimethylimidazilium chloride (Lepidiline D)	C_21_H_25_ClN_2_O	18.2	321	-	239	[43]
8	Campesterol	C_28_H_48_O	18.5	401	-	305, 190	[44]
9	9(S),12(S),13(S)-Trihydroxy-10(E)-octadecenoic acid (Pinellic acid)	C_18_H_34_O_5_	18.7	-	329	247, 151	[45]
10	1,3-dibenzyl-2-pentyl-4,5-dimethylimidazilium	C_24_H_31_N_2_^+^	19.6	347	-	309, 273, 239	[39,42]
11	N-octadecanamide	C_18_H_37_NO	20.9	284	-	239, 149, 121	[39]
12	N-benzyl-16-hydroxy-9-oxo-13E,15E-octadecadienamide	C_25_H_41_NO_2_	22.4	-	387	345, 247, 177	[39]

**Table 4 antioxidants-14-01214-t004:** Total phenolic content (TPC) and antioxidant activities of BM measured by CUPRAC, FRAP, ABTS, and DPPH assays.

Samples	TPC (mg GAE/mL)	CUPRAC (mg TEAC/mL)	FRAP (mg TEAC/mL)	ABTS IC_50_	DPPH IC_50_
BM	10.62 ± 0.56	14.06 ± 1.56	7.84 ± 0.49	15.26 ± 1.13	36.82 ± 9.57
Quercetin	1.23 ± 0.05	4.69 ± 0.24	3.14 ± 0.21	0.09 ± 0.03	0.09 ± 0.03
Trolox^®^	-	-	-	0.20 ± 0.06	0.18 ± 0.01

BM = aqueous extract of black maca; TPC = total phenolic content; CUPRAC = cupric reducing antioxidant capacity; FRAP = ferric-reducing antioxidant power; ABTS^•+^ = 2,2′-azinobis (3-ethylbenzothiazoline-6-sulfonic acid); DPPH = 2,2-diphenyl-1-picrylhydrazyl radical; GAE = Gallic acid equivalent; TEAC = Trolox^®^ equivalent antioxidant capacity. Results are expressed as means values ± SEM (*n* = 3).

**Table 5 antioxidants-14-01214-t005:** Free binding energy results (in kcal.mol^−1^) from molecular docking calculations of chemical constituents of black maca and reference compounds tramadol and estradiol with NAD(P)H oxidase (PDB ID: 2CDU), xanthine oxidase (PDB ID: 3NRZ, superoxide dismutase (PDB ID: 4MCM), μ-opioid receptor (PDB ID: 4DKL), and fatty acid amide hydrolase (PDB ID: 2VYA).

Compounds	Free Binding Energy (kcal.mol^−1^)				
	NAD(P)H Oxidase	Xanthine Oxidase	Superoxide Dismutase	μ-Opioid Receptor	Fatty Acid Amide Hydrolase
**1**	−6.1	−7.1	−7.0	−5.5	−7.1
**2**	−7.3	−6.9	−6.1	−6.9	−8.0
**3**	−7.9	−7.7	−6.6	−7.0	−8.3
**4**	−8.0	−7.9	−7.0	−7.5	−7.9
**5**	−8.8	−9.2	−6.7	−8.3	−9.0
**6**	−8.9	−9.3	−7.1	−8.4	−10.1
**7**	−8.7	−9.4	−7.1	−8.2	−9.3
**8**	−9.9	−9.5	−6.6	−8.6	−9.7
**9**	−6.5	−7.2	−6.2	−6.0	−7.4
**10**	−8.9	−9.6	−7.2	−7.7	−9.1
**11**	−6.4	−6.5	−5.3	−6.1	−7.1
**12**	−7.8	−8.4	−6.5	−7.4	−9.0
**tramadol**	ND	ND	ND	−6.4	ND
**estradiol**	ND	ND	ND	ND	−9.4

ND = not determined.

**Table 6 antioxidants-14-01214-t006:** ADMET properties of chemical constituents of black maca.

	Property									
	Absorption			Distribution			Metabolism	Excretion	Toxicity	
	Model Name									
Compounds	Caco-2	IA	SP	VD ss	BBB	CNS	CYP2D6/ CYP3A4 Inhibitor	TC	Oral Rat Acute Tox. (LD_50_)	Oral Rat Chronic Tox. -LOAEL
**1**	−0.288	2.571	−2.736	0.229	−1.697	−5.794	No/No	1.525	1.590	5.424
**2**	1.630	91.401	−2.206	0.507	0.481	−1.319	No/No	1.699	2.065	1.408
**3**	1.370	93.566	−2.597	0.591	−0.177	−1.716	No/No	1.834	2.109	2.564
**4**	1.496	94.101	−2.819	1.203	0.458	−1.619	No/No	0.986	2.912	0.897
**5**	1.470	94.519	−2.727	1.204	0.882	−1.213	Yes/No	0.935	2.604	0.205
**6**	1.557	94.681	−2.720	1.637	0.867	−1.197	Yes/No	0.893	0.565	0.159
**7**	1.477	95.526	−2.724	1.604	0.658	−1.305	Yes/Yes	0.908	2.780	0.063
**8**	1.235	95.601	−2.836	0.366	0.785	−1.835	No/No	0.572	2.206	0.935
**9**	0.861	38.813	−2.735	−0.882	−1.34	−3.314	No/No	1.967	2.897	1.188
**10**	0.265	89.944	−2.732	1.304	1.108	−1.161	Yes/No	1.123	2.527	0.056
**11**	1.560	89.664	−2.754	0.351	−.424	−1.704	No/No	1.906	1.786	0.903
**12**	0.411	87.941	−2.681	0.22	−0.633	−2.517	No/Yes	1.990	3.265	2.693

Caco-2: Caucasian colon adenocarcinoma permeability (Log Papp in 10^−6^cm/s); IA: Intestinal Absorption (% Absorbed); SP: Skin Permeability (log Kp); VDss: steady-state Volume of Distribution (Log L/kg); BBB: Blood–Brain Barrier permeability (Log BB); CNS: Central Nervous System (Log PS); CYP2D6: Cytochrome P450 2D6 inhibitor; CYP3A4: Cytochrome P450 3A4 inhibitor; TC: Total Clearance (Log mL/min/kg); LD_50_: Lethal Dose, 50% (mol/Kg); LOAEL: Lowest Observed Adverse Effect Level (Log mg/kg bw/day).

## Data Availability

Data are contained within the article.

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
