# Peer review of "Antioxidant, Neuroprotective, and Antinociceptive Effects of Peruvian Black Maca (Lepidium meyenii Walp.)"

_antioxidants, 2025, doi:10.3390/antiox14101214_

Round 1
Reviewer 1 Report
Overall, this study is well conceptualised supported with relevant experiments. However, the cohesiveness of the study can be improved by including a summary illustration depicting the postulated effects of each of the identified chemical constituent from the black maca extract.
Figures 5, 6, and 7. Please include the error bar for the dot-line plots.
Author Response
Manuscript ID: antioxidants-3888947
Reviewer 1
- Comment: Suggest to modify to: Antioxidant, Neuroprotective, and Antinociceptive Effects of Peruvian Black Maca (Lepidium meyenii) Majority of the results presented in this study focusses on the extract and not the individual identified phytochemicals. The original title might be misleading.
Answer: We thank the reviewer for this valuable suggestion. We agree that the original title could be misleading, as the study primarily focused on the extract rather than isolated metabolites. Accordingly, we agree the title proposed by the referee.
- Comment: Figures 5, 6, 7, for the dot-line plots, the error bars are missing.
Answer: We appreciate the reviewer’s observation. Error bars (mean ± SEM) have now been added to all dot-line plots in Figures 5–7, in accordance with the statistical methods described in the manuscript.
- Comment: Overall, this study is well conceptualised supported with relevant experiments. However, the cohesiveness of the study can be improved by including a summary illustration depicting the postulated effects of each of the identified chemical constituent from the black maca extract.
Answer: We sincerely thank the reviewer for this constructive suggestion. To improve the cohesiveness of the manuscript, we have included a graphical abstract that summarizes the identified metabolites (macamides, imidazole alkaloids, sterols, and fatty acid amides) along with their proposed molecular targets involved in redox regulation, nociception, and neuroprotection. This visual representation provides an integrated overview of the main findings.

Reviewer 2 Report
Comments
The manuscript with entitled “ Antioxidant, Neuroprotective, and Antinociceptive Effects of Peruvian Black Maca (Lepidium meyenii): Phytochemical Composition and Computational Analysis”.This article is meaningful, but requires major revisions.
- This manuscript discusses the antioxidant, neuroprotective, and antinociceptive role of Peruvian Black Maca (Lepidium meyenii). This topic is highly relevant and interest in health. It is intriguing. The manuscript is original.
- The introduction section is written in detail.
- In the methodology. â‘ All chemicals, reagents and kits should be in the list and provide source, concentration, the cargo number of the reagent.â‘¡What are total extraction rates for Peruvian Black Maca (Lepidium meyenii)? â‘¢In molecular docking analyses, how does the author evaluate that molecular docking has reached the basic level?
- In the results, since the author conducted animal experiments, the antioxidant capacity indicators in brain tissue should be measured.
- The Latin name format of the species should meet the requirements.
- In the conclusions, the limitations of the research and the introduction of future research directions are well presented.
- There are some grammar and style errors, especially, the capitalization of words.
Author Response
Manuscript ID: antioxidants-3888947
Reviewer 2
- Comments: This manuscript discusses the antioxidant, neuroprotective, and antinociceptive role of Peruvian Black Maca (Lepidium meyenii). This topic is highly relevant and interest in health. It is intriguing. The manuscript is original.
Answer: We thank the reviewer for recognizing the relevance, originality, and scientific value of our study on the antioxidant, neuroprotective, and antinociceptive effects of Peruvian Black Maca (Lepidium meyenii). We sincerely appreciate the positive evaluation.
- 2. Comments: The introduction section is written in detail.
Answer: We thank the reviewer for this encouraging remark. We are pleased that the Introduction was considered detailed and appropriate.
- 3. In the methodology.
Comments:
â‘ All chemicals, reagents and kits should be in the list and provide source, concentration, the cargo number of the reagent.
Answer: We thank the reviewer for this comment. We have expanded the Materials and Methods section to include of chemicals, reagents, and kits, in accordance with the Antioxidants journal guidelines.
â‘¡What are total extraction rates for Peruvian Black Maca (Lepidium meyenii)?
Answer: We thank the reviewer for their questions. From 500 g of dried hypocotyls, we obtained 72.5 g of lyophilized extract (yield: 14.5% w/w). This information has been added to the revised manuscript.
â‘¢In molecular docking analyses, how does the author evaluate that molecular docking has reached the basic level?
Answer: We thank the reviewer for this question. To ensure that our molecular docking analysis achieved a basic and reliable level of validation, we implemented several complementary procedures. First, we performed re-docking of the co-crystallized ligands into their respective protein binding sites using the same docking parameters. The resulting root-mean-square deviation (RMSD) values between the re-docked and crystallographic poses were consistently below 2.0 Å, confirming accurate reproduction of the experimentally observed binding modes and meeting the widely accepted validation threshold [1,2]. In addition, we carried out at least 10 independent docking runs for each ligand–protein system, which allowed us to verify the reproducibility of the predicted binding poses and the robustness of the calculated binding energies. Collectively, these validation steps demonstrate that our docking protocol is both reliable and consistent, in accordance with established validations standards reported in the field [1,2]. This information has now been explicitly into the revised manuscript and supplementary material.
References:
[1] Hevener KE, Zhao W, Ball DM, Babaoglu K, Qi J, White SW, Lee RE. Validation of molecular docking programs for virtual screening against dihydropteroate synthase. J Chem Inf Model. 2009, 49(2), 444–460. doi:10.1021/ci800293n
[2] Mateev E, Valkova I, Angelov B, Georgieva M, Zlatkov A. Validation through re-docking, cross-docking and ligand enrichment in various well-resoluted MAO-B receptors. Int J Pharm Sci Res. 2021, 13, 1099. doi:10.13040/IJPSR.0975-8232.13(3).1099-07
- Comments: In the results, since the author conducted animal experiments, the antioxidant capacity indicators in brain tissue should be measured.
Answer: We thank the reviewer for this valuable comment. In addition to the in vitro antioxidant capacity assays performed with the aqueous extract of black maca, we measured malondialdehyde (MDA) levels in brain tissue during the in vivo experiments. MDA is a well-established biomarker of lipid peroxidation and oxidative stress. These data, presented in Figure 4, confirm that Peruvian Black Maca exhibits antioxidant effects both in vitro and in vivo. We have revised the Results and Discussion sections to highlight this more clearly.
- Comments: The Latin name format of the species should meet the requirements.
Answer: We thank the reviewer for this observation. We have carefully revised the manuscript to ensure that all Latin expressions (in vivo, in vitro, in silico, ad libitum, etc.) and scientific names comply with the journal’s formatting requirements. All corrections have been highlighted in the revised version of the manuscript.
- Comments: In the conclusions, the limitations of the research and the introduction of future research directions are well presented.
Answer: We appreciate this positive feedback. We are glad that the conclusions, including the identified limitations and perspectives for future research, were considered clear and well articulated.
- Comments: There are some grammar and style errors, especially, the capitalization of words.
Answer: We thank the reviewer for pointing this out. The manuscript has been thoroughly revised for grammar, style, and capitalization to ensure consistency and fluency. All changes have been highlighted in the revised version of the manuscript.

Reviewer 3 Report
This manuscript describes the phytochemical profile and evaluates the antioxidant, antinociceptive, and neuroprotective properties of a lyophilized aqueous extract of BM hypocotyls. The organization and English are fine. Presentation quality is very acceptable. Major concerns would be missing a few controls in some of the related bioassays which should come from the bioactive compounds identified from the molecular modeling studies.
A few concerns for the authors.
- On page 3, 1st paragraph: what is the total weight of the lyophilized sample obtained from 500 g of dried hypocotyls? What is the yield?
- On page 6, session 3.1. Chemical Composition of lyophilized hypocotyl aqueous extract of black maca: What are the AUCs and percentages of the 12 samples identified from the lyophilized hypocotyl aqueous extract of black maca? From the Figure S1, there are other peaks which are comparable to some of the identified compounds, any idea what they are? How can the authors conclude that those unidentified compounds are not contribute to the bioactivities as the molecular modeling studies indicate?
- On page 15, line 452-454: “The docking interactions (Figure 8) illustrate the binding modes of lepidiline B (compound 6) and campesterol (compound 8) with key protein targets related to redox homeostasis, nociception, and neuroprotection.” It would be very helpful and confirming to have these two compounds tested together with the lyophilized hypocotyl aqueous extract of black maca in
Author Response
Manuscript ID: antioxidants-3888947
Reviewer 3
1.- Comment: Missing some controls in some bioassays. Methods need more details.
Answer: We thank the reviewer for this important observation. While our study included positive controls (tramadol in the nociceptive assays and estradiol in the OVX model), we acknowledge that isolated metabolites (e.g., lepidiline B, campesterol) were not tested in parallel. We now state this as a limitation in the Discussion and have indicated that future studies should incorporate isolated metabolites to better validate their contribution to the observed bioactivities.
2.- Comment: On page 3, 1st paragraph: what is the total weight of the lyophilized sample obtained from 500 g of dried hypocotyls? What is the yield?
Answer: We thank the reviewer for such questions. This information is now included in the revised manuscript. From 500 g of dried hypocotyls, we obtained 72.5 g of lyophilized extract, corresponding to a yield of 14.5% (w/w).
3.- Comment: On page 6, session 3.1. Chemical Composition of lyophilized hypocotyl aqueous extract of black maca: What are the AUCs and percentages of the 12 samples identified from the lyophilized hypocotyl aqueous extract of black maca? From the Figure S1, there are other peaks which are comparable to some of the identified compounds, any idea what they are? How can the authors conclude that those unidentified compounds are not contribute to the bioactivities as the molecular modeling studies indicate?
Answer: We appreciate this insightful comment. In the present study, relative abundances (AUC percentages) for the identified compounds were not determined, and therefore we cannot provide quantitative contributions of each metabolite. Moreover, some additional chromatographic peaks remained unidentified with the available standards and databases. We cannot exclude the possibility that these unidentified compounds may contribute to the observed biological activities, as also suggested by the docking analyses. We have added a statement in the Discussion acknowledging this limitation and emphasizing that future studies will focus on isolating and structurally characterizing these unidentified metabolites to clarify their potential bioactive roles.
4.- Comment: On page 15, line 452-454: “The docking interactions (Figure 8) illustrate the binding modes of lepidiline B (compound 6) and campesterol (compound 8) with key protein targets related to redox homeostasis, nociception, and neuroprotection.” It would be very helpful and confirming to have these two compounds tested together with the lyophilized hypocotyl aqueous extract of black maca in
Answer: We thank the reviewer for this constructive suggestion. We fully agree that testing lepidiline B (compound 6) and campesterol (compound 8) alongside the lyophilized aqueous extract would provide stronger evidence to confirm their contribution. However, due to limitations in compound isolation and availability, this was not feasible within the scope of the present study. Our primary objective was to evaluate the biological activity of the whole extract, as traditionally consumed. The need for subsequent studies focusing on the isolation and experimental evaluation of these bioactive compounds has now been emphasized in the Discussion and in the Conclusions/Future Perspectives section. This applies to their assessment both individually and in combination with the extract, with the aim of further validating the molecular docking results.

Round 2
Reviewer 2 Report
None None